# Synergistic application of artificial intelligence and response surface methodology for predicting and enhancing in vitro tuber production of potato (*Solanum tuberosum*)

Rajermani Thinakaran[1], Ecenur Korkmaz[2], Başak Ünver[2], Seyid Amjad Ali[3], Zeshan Iqbal[4], Muhammad Aasim[5]*

**1** Faculty of Data Science and Information Technology, INTI International University, Negeri Sembilan, Malaysia, **2** Öztar Tohumculuk ve Tarım Ürünleri A.Ş. Izmir, Türkiye, **3** Department of of Information Systems and Technologies, Bilkent University, Ankara, Türkiye, **4** Department of Computer Engineering, Sivas University of Science and Technology, Sivas, Türkiye, **5** Faculty of Agricultural Sciences and Technologies, Sivas University of Science and Technology, Sivas, Türkiye

* mshazim@gmail.com

## Abstract

In vitro regeneration of potato tubers is highly significant in modern agriculture as it offers efficient propagation, genetic enhancement, and pathogen-free seed production. This study aimed to optimize in vitro tuberization by manipulating key variables, including cultivar, sucrose concentration, and cytokinin-auxin interactions. Results were analyzed by response surface regression analysis (RSRA) of Response Surface Methodology (RSM), followed by data validation and prediction with machine learning (ML) models. Fontana cultivar exhibited superior tuberization performance, with a maximum tuberization rate of 75.6% from Murashige and Skoog (MS) medium supplemented with 90 g/L sucrose, 2 mg/L BAP, and 1 mg/L Indole-3-butyric acid (IBA). Sucrose concentration was the most significant factor for all growth parameters, particularly tuber size and weight. RSRA analysis confirmed the significance of the linear effects of sucrose and BAP on tuberization, while auxins primarily regulated tuber size and weight. Pareto chart analysis highlighted sucrose as the most influential variable for both cultivars. Heatmap and network plot analyses further illustrated strong positive correlations between sucrose, BAP, and tuber formation, whereas auxins exhibited comparatively weaker effects. Results analyzed by Machine learning (ML) models revealed maximum predictive accuracy for tuberization by Random Forest (RF) model with an $R^2$ of 0.379. However, all other models also faced challenges with high error rates, indicating the need for improved feature engineering. This study concludes that optimizing sucrose concentration and BAP levels, combined with selective auxin application, and integration of RSM and AI presents a promising strategy for optimization and potentially improving large-scale commercial production of disease-free potato tubers.

**Data availability statement:** The data and the code used in this article can be accessed under GPL-3.0 at Zenodo: https://doi.org/10.5281/zenodo.15516978

**Funding:** The author(s) received no specific funding for this work.

**Competing interests:** The authors have declared that no competing interests exist.

## Introduction

In vitro regeneration of potato tubers is a valuable technique for research and commercial agriculture due to its potential for enhancing propagation, genetic improvement, and pathogen-free production [1]. This method allows for the rapid multiplication of plants under controlled conditions, bypassing environmental limitations and ensuring year-round cultivation. It also preserves genetic resources, allowing researchers to introduce desirable traits like disease resistance or enhanced yield into new cultivars [2]. In vitro regeneration is a cornerstone in modern potato breeding and crop management programs. In vitro regeneration of potato tubers presents several challenges, including optimal environmental conditions, hormonal imbalances, somaclonal variation, microbial contamination, and acclimatization to external conditions [3]. These factors can affect the efficiency and quality of tuber formation, leading to inconsistent tuber size and quality. Genetic mutations during tissue culture can lead to undesirable traits or reduced uniformity among regenerated plants [4]. Microbial contamination is a persistent problem, as in vitro systems are sensitive to pathogens. Proper acclimatization to external conditions can also be challenging, reducing survival rates when transferred to the field [5].

In vitro production of potato tubers can be improved by optimizing culture conditions, managing growth regulators, reducing somaclonal variation, improving microbial control, and implementing gradual acclimatization techniques. Proper management of growth hormones, such as cytokinins and auxins, is crucial to avoid hormonal imbalances and improve tuber quality and uniformity [6]. Similarly, genotype selection can be done to select genotypes with superior tuberization potential and introduce traits that improve in vitro performance [7]. Sucrose concentration is crucial for potato tuberization, influencing induction rate and tuber size. It triggers formation, promotes enlargement, and increases starch accumulation [8]. Sucrose works with hormones to regulate tuberization, enhancing cytokinin effects and reducing gibberellin inhibitory effects [4]. By addressing these factors, the efficiency and reliability of potato tuber production can be significantly improved, leading to higher-quality plantlets and increased scalability for commercial and research purposes.

Optimization of balance hormone levels and sucrose concentration for a specific cultivar is vital for efficient and high-quality tuber production in vitro. Optimization of these variables can be attained by employing modern statistical tools like Response Surface Methodology (RSM). It helps in understanding the interactions between various factors, optimizing their levels, and improving the efficiency of experimental designs [9]. It can help in determining the optimal concentration of growth hormones, and sucrose concentration for efficient tuber induction and growth [9] as targeted in this study.

Artificial Intelligence (AI) is revolutionizing the in vitro culture due to enhanced efficiency, precision, scalability, development of predictive models, and automation of tissue culture processes by monitoring and controlling multiple factors in real time [6]. It is possible to analyze large datasets to predict optimal combinations of growth factors like hormones, sucrose, temperature, photoperiod, etc. to maximize tuber's yield

and quality by employing AI models [8]. In recent years, the use of AI-based models in plant tissue culture has been documented for different commercial crops for precise prediction, validation, or optimization of protocols [9]. The integration of machine learning (ML) algorithms with RSM is a revolutionary approach to solving complex biological problems [8]. It is possible to analyze big data, uncovering hidden parameters and predictive insights complemented by the RSM. ML tools like support vector machines (SVMs), decision trees, and neural networks can identify non-linear relationships between variables, and handle high-dimensional datasets and complex interactions [8]. This synergy between RSM and ML is highly efficient for generating predictive models for experiments scaling up from laboratory to commercial applications [10]. In this study, optimization of in vitro potato tuberization by manipulating multiple factors like cultivar (cv), cytokinin-auxin interactions, and sucrose concentration was conducted. The study also aims to investigate the synergistic benefits of integrating RSM and AI to enhance efficiency and reproducibility, facilitating successful protocols for commercial applications.

## Materials and methods

### In vitro tuberization

In this study, two commercial potato varieties (Fontane and Agria), which are commonly used for making potato chips due to their high starch contents were used. The potato tubers (seed) were procured from Öztar Tohumculuk ve Tarım Ürünleri A.Ş., Türkiye. Tubers were stored in the dark for one month for sprouting, and these sprouts were surface sterilization. Thereafter, the meristem culture technique was used for in vitro microtuber production. Half-strength Murashige and Skoog (MS) medium (2.2 g/L) was used as a standard nutrient media. Commercial sugar was used to ensure carbon sources at three different doses of 30, 60, and 90 mg/L. To investigate the impact of different plant growth regulators (PGRs), Benzylaminopurine (BAP) was applied at the rate of 2.0 mg/L with different auxins like Indole-3-butyric acid (IBA), 1-Naphthaleneacetic acid (NAA), and Indole-3-acetic acid (IAA). The combinations of PGRs used in this study were comprised of; (i) 2.0 mg/L BAP alone (control treatment with cytokinin), (ii) 2.0 mg/L BAP + 1.0 mg/L IBA, (iii) 2.0 mg/L BAP + 1.0 mg/L NAA, and (iv) 2.0 mg/L BAP + 1.0 mg/L IAA with three different sugar concentrations. Additionally, a control experiment was performed by using a nutrient medium without PGRs but enriched with sugar only.

A total of 15 different combinations were used to investigate the best medium for both varieties. The experimental layout was a completely randomized design (CRD) with three replications and five explants per replication. The experiment was repeated thrice and an average of all experiments was used for statistical analysis. Data collection and observations were recorded at regular intervals. For data analysis, three different strategies were employed for data validation and optimization. In the first step, one-way analysis of variance (ANOVA – Table 1) and interaction (Table 2) was investigated to find

**Table 1. One-way ANOVA analysis of cultivar, BAP, and sucrose on in vitro tuberization of potato.**

| Cultivar | Tuberization (%) | Tubers per plant | Tuber size (cm$^2$) | Tuber weight (g) |
|---|---|---|---|---|
| Agria | $36.52 \pm 4.02^{ns}$ | $0.640 \pm 0.06^{ns}$ | $0.293 \pm 0.03^{ns}$ | $0.120 \pm 0.01^{ns}$ |
| Fontane | $31.48 \pm 4.23$ | $0.613 \pm 0.06$ | $0.375 \pm 0.04$ | $0.162 \pm 0.02$ |
| p-value | 0.389 | 0.735 | 0.102 | 0.056 |
| **BAP** | **Tuberization (%)** | **Tubers per plant** | **Tuber size (cm$^2$)** | **Tuber weight (g)** |
| 0 | $12.41 \pm 4.19b$ | $0.333 \pm 0.09b$ | $0.147 \pm 0.04b$ | $0.060 \pm 0.02b$ |
| 2 | $39.40 \pm 3.32a$ | $0.699 \pm 0.04a$ | $0.381 \pm 0.03a$ | $0.161 \pm 0.01a$ |
| p-value | 0.000** | 0.000** | 0.000** | 0.000** |
| **Sucrose** | **Tuberization (%)** | **Tubers per plant** | **Tuber size (cm$^2$)** | **Tuber weight (g)** |
| 30 | $7.89 \pm 1.70c$ | $0.440 \pm 0.07b$ | $0.176 \pm 0.03b$ | $0.074 \pm 0.02b$ |
| 60 | $35.67 \pm 4.66b$ | $0.640 \pm 0.07ab$ | $0.377 \pm 0.05a$ | $0.151 \pm 0.02a$ |
| 90 | $58.44 \pm 5.16a$ | $0.799 \pm 0.06a$ | $0.449 \pm 0.04a$ | $0.198 \pm 0.02a$ |
| p-value | 0.000** | 0.001** | 0.000** | 0.000** |

**Table 2. One way ANOVA analysis of interaction of BAP, sugar, and different auxins on in vitro tuberization of potato.**

| Int. | Cv. | Suc (g/L) | BAP (mg/L) | IBA (mg/L) | NAA (mg/L) | IAA (mg/L) | Tuberization (%) | Tubers per plant | Tuber size (cm²) | Tuber weight (g) |
|---|---|---|---|---|---|---|---|---|---|---|
| 1 | Agria | 30 | 0 | – | – | – | 0.00±0.00d | 0.00±0.00b | 0.00±0.00b | 0.00±0.00e |
| 2 | Agria | 30 | 2 | – | – | – | 23.33±7.33abcd | 0.80±0.20a | 0.191±0.05ab | 0.079±0.03bcde |
| 3 | Agria | 30 | 2 | 1 | – | – | 6.67±3.24bcd | 0.60±0.25ab | 0.261±0.12ab | 0.108±0.05abcde |
| 4 | Agria | 30 | 2 | – | 1 | – | 13.33±6.48abcd | 0.60±0.25ab | 0.153±0.07ab | 0.051±0.03cde |
| 5 | Agria | 30 | 2 | – | – | 1 | 15.56±9.53abcd | 0.40±0.20ab | 0.095±0.06ab | 0.030±0.02de |
| 6 | Agria | 60 | 0 | – | – | – | 3.33±3.33 cd | 0.20±0.20ab | 0.051±0.05ab | 0.015±0.01de |
| 7 | Agria | 60 | 2 | – | – | – | 53.30±14.14abcd | 0.80±0.20a | 0.284±0.08ab | 0.104±0.03abcde |
| 8 | Agria | 60 | 2 | 1 | – | – | 61.10±16.19abcd | 0.80±0.20a | 0.452±0.13ab | 0.177±0.05abcde |
| 9 | Agria | 60 | 2 | – | 1 | – | 65.60±17.50abcd | 0.80±0.20a | 0.519±0.13ab | 0.191±a0.05bcd |
| 10 | Agria | 60 | 2 | – | – | 1 | 38.90±16.10abcd | 0.60±0.25ab | 0.313±0.13ab | 0.114±0.05abcde |
| 11 | Agria | 90 | 0 | – | – | – | 33.33±14.18abcd | 0.80±0.20a | 0.35±00.09ab | 0.148±0.05abcde |
| 12 | Agria | 90 | 2 | – | – | – | 52.20±13.46abcd | 0.80±0.20a | 0.350±0.09ab | 0.146±0.04abcde |
| 13 | Agria | 90 | 2 | 1 | – | – | 62.20±16.05abcd | 0.80±0.20a | 0.484±0.13ab | 0.224±0.06abc |
| 14 | Agria | 90 | 2 | – | 1 | – | 68.90±17.80abc | 0.80±0.20a | 0.492±0.13ab | 0.234±0.06ab |
| 15 | Agria | 90 | 2 | – | – | 1 | 50.00±14.04abcd | 0.80±0.20a | 0.408±0.10ab | 0.183±0.05abcd |
| 16 | Fontane | 30 | 0 | – | – | – | 0.00±0.00d | 0.00±0.00b | 0.000±0.00b | 0.000±0.000e |
| 17 | Fontane | 30 | 2 | – | – | – | 0.00±0.00d | 0.00±0.00b | 0.000±0.00b | 0.000±0.00e |
| 18 | Fontane | 30 | 2 | 1 | – | – | 2.22±1.36 cd | 0.40±0.25ab | 0.240±0.15ab | 0.110±0.06abcde |
| 19 | Fontane | 30 | 2 | – | 1 | – | 7.78±2.83bcd | 0.80±0.20a | 0.450±0.15ab | 0.241±0.08ab |
| 20 | Fontane | 30 | 2 | – | – | 1 | 10.00±3.24abcd | 0.80±0.20a | 0.366±0.15ab | 0.119±0.04abcde |
| 21 | Fontane | 60 | 0 | – | – | – | 4.44±2.73 cd | 0.40±0.25ab | 0.204±0.31ab | 0.083±0.06abcde |
| 22 | Fontane | 60 | 2 | – | – | – | 25.60±12.61abcd | 0.60±0.25ab | 0.264±0.13ab | 0.103±0.05abcde |
| 23 | Fontane | 60 | 2 | 1 | – | – | 35.60±12.25abcd | 0.80±0.20a | 0.616±0.16ab | 0.254±0.07ab |
| 24 | Fontane | 60 | 2 | – | 1 | – | 42.20±12.25abcd | 0.80±0.20a | 0.591±0.15ab | 0.249±0.06ab |
| 25 | Fontane | 60 | 2 | – | – | 1 | 26.70±11.45abcd | 0.60±0.25ab | 0.477±0.20ab | 0.224±0.10abc |
| 26 | Fontane | 90 | 0 | – | – | – | 33.30±14.58abcd | 0.60±0.25ab | 0.278±0.11ab | 0.114±0.05abcde |
| 27 | Fontane | 90 | 2 | – | – | – | 73.3±18.690ab | 0.87±0.23a | 0.491±0.13ab | 0.210±0.06abc |
| 28 | Fontane | 90 | 2 | 1 | – | – | 75.60±19.36a | 0.89±0.23a | 0.675±0.17a | 0.260±0..07a |
| 29 | Fontane | 90 | 2 | – | 1 | – | 67.80±17.17abc | 0.81±0.20a | 0.527±0.14ab | 0.253±0.07ab |
| 30 | Fontane | 90 | 2 | – | – | 1 | 67.80±18.21abc | 0.82±0.20a | 0.442±0.12ab | 0.210±0..06abc |
| | | | | | | p-value | 0.000** | 0.026* | 0.000** | 0.000** |

**significant at p0.001;

*significant at p0.005; Int: interaction, Suc: sucrose.

out the best possible treatments. In the second step, data was subjected to response surface regression analysis (RSRA) for the whole experiment (Table 3), followed by analysis for the individual variety (Table 4). The RSRA was enriched with a Pareto chart and Normal plot analysis for optimization. Data was also analyzed with heatmap and network plot analysis for individual potato varieties. In the last step, data was validated and predicted by machine learning (ML) analysis.

## Machine learning analysis

For ML analysis, three different models (Random Forest – RF, Support Vector Regressor – SVR, and Light Gradient Boosting Machine – LightGBM) were used for validation and prediction of outcomes of in vitro regeneration. These models were chosen based on their ability to handle non-linear relationships, structured data, and complex predictive tasks. In

**Table 3. Combined response surface regression analysis of BAP, sugar, and different auxins on in vitro tuberization of potato.**

| Source | Tuberization (%) | | Tubers per plant | | Tuber size (cm²) | | Tuber weight (g) | |
|---|---|---|---|---|---|---|---|---|
| | F-Value | P-Value | F-Value | P-Value | F-Value | P-Value | F-Value | P-Value |
| Model | 7.44 | 0.000** | 2.63 | 0.001** | 4.19 | 0.000** | 4.50 | 0.000** |
| Linear | 5.56 | 0.000** | 3.41 | 0.004** | 5.81 | 0.000** | 6.30 | 0.000** |
| BAP | 13.01 | 0.000** | 7.01 | 0.009** | 2.95 | 0.088 | 2.54 | 0.114 |
| Sucrose | 8.21 | 0.005** | 0.74 | 0.392 | 1.42 | 0.235 | 2.02 | 0.158 |
| IBA | 0.13 | 0.715 | 0.36 | 0.550 | 7.99 | 0.005** | 7.79 | 0.006** |
| NAA | 0.79 | 0.376 | 1.12 | 0.292 | 8.04 | 0.005** | 10.73 | 0.001** |
| IAA | 0.20 | 0.658 | 0.05 | 0.826 | 1.64 | 0.202 | 1.84 | 0.177 |
| Cultivar | 0.03 | 0.865 | 2.87 | 0.092 | 3.10 | 0.081 | 3.60 | 0.060 |
| Square | 0.28 | 0.600 | 0.07 | 0.795 | 2.01 | 0.159 | 0.62 | 0.432 |
| Sucrose*Sucrose | 0.28 | 0.600 | 0.07 | 0.795 | 2.01 | 0.159 | 0.62 | 0.432 |
| 2-Way Interaction | 0.83 | 0.591 | 1.06 | 0.398 | 0.43 | 0.919 | 0.48 | 0.888 |
| BAP*Sucrose | 1.05 | 0.308 | 0.86 | 0.356 | 0.00 | 0.946 | 0.01 | 0.910 |
| BAP*Cultivar | 0.54 | 0.466 | 1.75 | 0.188 | 0.14 | 0.709 | 0.07 | 0.785 |
| Sucrose*IBA | 0.59 | 0.444 | 0.10 | 0.758 | 0.00 | 0.982 | 0.01 | 0.929 |
| Sucrose*NAA | 0.15 | 0.701 | 1.29 | 0.257 | 0.50 | 0.482 | 0.34 | 0.563 |
| Sucrose*IAA | 0.08 | 0.774 | 0.60 | 0.441 | 0.62 | 0.434 | 0.06 | 0.810 |
| Sucrose*Cultivar | 2.69 | 0.103 | 0.18 | 0.669 | 0.00 | 0.960 | 0.16 | 0.690 |
| IBA*Cultivar | 0.10 | 0.754 | 1.36 | 0.246 | 0.99 | 0.321 | 0.55 | 0.461 |
| NAA*Cultivar | 0.00 | 1.000 | 2.64 | 0.107 | 1.36 | 0.245 | 2.52 | 0.115 |
| IAA*Cultivar | 0.50 | 0.482 | 3.70 | 0.057 | 1.76 | 0.186 | 1.87 | 0.174 |
| Lack-of-Fit | 0.88 | 0.574 | 0.65 | 0.805 | 0.63 | 0.826 | 0.63 | 0.829 |

**significant at p0.001;

*significant at p0.005.

this study, the leave-one-out cross-validation (LOO-CV) method [11] was used to evaluate the performance of all tested models. The data was divided into two sets of training and testing, as the number of instances as folds remained the same in the dataset in the LOO-CV technique. Whereas training and testing are performed with the remaining instances and the selected instances during the learning process. A grid search technique was used to find the optimal hyperparameters for seeking the best model. Whereas the Sklearn package [12] and the open-source Python programming language [13] were used for coding. The performance of the models was evaluated with the help of four different performance metrics.

With the help of bagging, also known as bootstrap aggregation [14], the RF model is an ensemble learning algorithm that is based on multiple decision trees [15] and aggregates their predictions to improve accuracy and reduce overfitting. The RF model is efficient, particularly for handling non-linear patterns and high-dimensional datasets, which makes the model suitable for analyzing complex biological processes like tuberization.

$$y = \sum_{i=1}^{n} (\alpha_i - \alpha_i^*) k(x, x_i) + b \tag{1}$$

$y$ = data point value; $n$ = sampling size (number)

The SVR model is a kernel-based learning method based on constructing input features into a higher-dimensional space to capture complex relationships [16]. The SVR model is based on minimizing error while maintaining model generalization and is highly suitable for datasets that exhibit heterogeneity in variance.

**Table 4. Response surface regression analysis of BAP, sugar, and different auxins on in vitro tuberization of potatoes of individual variety.**

| Source | Fontana | | | | Agria | | | |
|---|---|---|---|---|---|---|---|---|
| | Tuberiza-tion (%) | Tubers per plant | Tuber size (cm2) | Tuber weight (g) | Tuberia-tion (%) | Tubers per plant | Tuber size (cm2) | Tuber weight (g) |
| Model | 0.000** | 0.007** | 0.002** | 0.001** | 0.000** | 0.037** | 0.001** | 0.000** |
| Linear | 0.034* | 0.052* | 0.007** | 0.003** | 0.002** | 0.022* | 0.003** | 0.001** |
| BAP | 0.038* | 0.353 | 0.408 | 0.420 | 0.004** | 0.006** | 0.080 | 0.111 |
| Sucrose | 0.104 | 0.456 | 0.830 | 0.818 | 0.020* | 0.051* | 0.023* | 0.007** |
| IBA | 0.618 | 0.216 | 0.021* | 0.034* | 0.972 | 0.687 | 0.124 | 0.080 |
| NAA | 0.515 | 0.062 | 0.016* | 0.004** | 0.545 | 0.687 | 0.161 | 0.148 |
| IAA | 0.848 | 0.134 | 0.111 | 0.099 | 0.434 | 0.229 | 0.969 | 0.994 |
| Square | 0.289 | 0.714 | 0.261 | 0.339 | 0.091 | 1.000 | 0.398 | 0.993 |
| Sucrose*Sucrose | 0.289 | 0.714 | 0.261 | 0.339 | 0.091 | 1.000 | 0.398 | 0.993 |
| 2-Way Interaction | 0.433 | 0.164 | 0.387 | 0.540 | 0.719 | 0.349 | 0.837 | 0.698 |
| BAP*Sucrose | 0.094 | 0.515 | 0.429 | 0.413 | 0.861 | 0.051* | 0.333 | 0.338 |
| Sucrose*IBA | 1.000 | 0.357 | 0.835 | 0.609 | 0.296 | 0.621 | 0.747 | 0.571 |
| Sucrose*NAA | 0.573 | 0.040* | 0.127 | 0.095 | 0.296 | 0.621 | 0.360 | 0.172 |
| Sucrose*IAA | 0.511 | 0.042* | 0.126 | 0.312 | 0.827 | 0.325 | 0.432 | 0.311 |
| Lack-of-Fit | 0.910 | 0.822 | 0.984 | 0.941 | 0.361 | 0.918 | 0.447 | 0.688 |

**significant at p0.001;

*significant at p0.005.

LightGBM is a gradient-boosting framework model based on histogram-based learning to speed up computation and accurate prediction. LightGBM model is particularly useful for handling imbalanced data and high-dimensional feature spaces, enabling the model to employ biological research for predictive modeling [17].

## Performance metrics

The performance of each model was performed with the aid of five different performance metrics.

The coefficient of determination ($R^2$) is used for quantifying the proportion of variance in the dependent variable. A higher $R^2$ score indicates a better fit between predicted and actual values, and its value ranges from 0 to 1.

$$R^2 = 1 - \frac{\sum_{i=1}^{n} (Y_i - \hat{Y}_i)^2}{\sum_{i=1}^{n} (Y_i - \widetilde{Y})^2}$$

(2)

Root Mean Squared Error (RMSE) measures the square root of the average squared differences between predicted and actual values. The better performance of the model using this metric is based on a low RMSE score.

$$RMSE = \sqrt{\frac{1}{n} \sum_{i=1}^{n} (Y_i - \hat{Y}_i)^2}$$

(3)

Mean Absolute Error (MAE) measures the absolute differences between predicted and actual values, and a low MAE score refers to better performance of the model.

$$MAE = \frac{1}{n} \sum_{i=1}^{n} \left| Y_i - \hat{Y}_i \right|$$

(4)

Median Absolute Error (MedAE) is the median of the absolute errors between predicted and actual values. MedAE is relatively less sensitive to outliers and extreme deviations, making it suitable for complex biological data. Likewise, in RMSE and MAE, a lower MedAE score refers to greater model robustness and consistency.

$$MedAE = median \left( \left| Y_1 - \hat{Y}_1 \right|, \ldots, \left| Y_n - \hat{Y}_n \right| \right)$$

(5)

Mean Logarithmic Squared Error (MLSE) measures the relative prediction errors, accounting for logarithmic residual transformations. A lower MLSE value presents the reduced absolute deviation between predicted and actual values.

$$MSLE = \frac{1}{n} \sum_{i=1}^{n} \left( \log (Y_i + 1) - \log \left( \hat{Y}_i + 1 \right) \right)^2$$

(6)

Where, $Y_i$ = measured value; $\hat{Y}_i$ = predicted value; $\overline{Y}$ = measured value's mean; $n$ = count of samples

## Results

The study examines the impact of plant growth regulators and sucrose on in vitro potato tuberization, analyzing data on BAP concentration, sucrose levels, and auxin types across two potato cultivars. Results show significant variation among treatments and cultivars, providing a comprehensive understanding of how specific hormonal and carbohydrate inputs influence tuber initiation and development.

### Impact of BAP, sucrose and auxins

A comparison of cultivars is presented in Table 1 which revealed a statistically insignificant impact for all variables. However, Fontance cv. was superior in terms of tuberaization, tubers per plant, size, and weight compared to Agria cv. Investigating the impact of BAP concentration alone illustrated the significant impact of BAP concentration and supplementation of 2.0 mg/L BAP resulted in more tuber production with more size and weight. Supplementation of sucrose concentration regulated the in vitro tuber production statistically significant, and all variables increased with an increase of sucrose concentration respectively.

The interaction values offer additional insights into the synergistic effects of these variables on tuberization outcomes, facilitating the identification of ideal conditions for potato tuber development. The interaction between sucrose and BAP significantly influences the percentage of tuberization. A culture medium consisting of 90 g/L sucrose, 2 mg/L BAP, and 1.0 mg/L IBA produced the highest tuberization percentage (75.6%) for Fontane cv. Conversely, no tubers were induced from Agria cv. Cultured on medium provided with 30 g/L sucrose and 0 mg/L BAP. Whereas supplementation of 90 g/L sucrose, 2 mg/L BAP, and 1.0 mg/L NAA yielded 68.9% for Agria cv. and 67.8% for Fontane cv. when cultured on medium provided with 90 g/L sucrose, 2 mg/L BAP, and 1.0 mg/L IAA. The number of tubers per plant was similarly affected by the culture medium. Supplementation of 90 g/L sucrose with 2 mg/L BAP yielded maximum tubers per plant (0.86) for Fontanne cv. In general, relatively more tubers per plant were generated from Fontanne cv. Compared to Agria cv.

The results shown in Table 2 illustrate the impact of the variables, sucrose and BAP, on the primary outcomes of tuber size and weight, with a specific focus on their interactions.

The size of tubers is significantly influenced by PGRs and supplementation of 90 g/L sucrose, 2 mg/L BAP, and 1.0 mg/L IBA produced largest tubers with average size of 0.675 cm² followed by 0.616 cm² from 60 g/L sucrose, 2 mg/L BAP, and 1.0 mg/L IBA and 0.591 cm² form 60 g/L sucrose, 2 mg/L BAP, and 1.0 mg/L NAA from Fontanne cv. On the contrary, relatively smaller-sized tubers were recorded for Agria cv. with the largest tubers attained from medium containing 60 g/L sucrose, 2 mg/L BAP, and 1.0 mg/L NAA. The weight of the tuber exhibited a trend analogous to that of its size, and most

heaviest tubers (0.260 g) were obtained from the combination of 90 g/L sucrose, 2 mg/L BAP, and 1.0 mg/L IBA. Results revealed the significant impact of sucrose concentration on tuber size and weight.

## Response surface regression analysis

This study assessed the importance of critical variables for in vitro potato tuber formation utilizing Response Surface Methodology (RSM), emphasizing response surface regression analysis. The results indicated that multiple factors significantly influence tuberization percentage, tuber count per plant, tuber size, and tuber weight. The model demonstrated strong significance across all parameters, with P-values below 0.05, so affirming the validity of the response surface model employed for optimization. BAP (F = 13.01, P = 0.000) and sucrose content (F = 8.21, P = 0.005) were determined to be significant factors influencing tuberization percentage. These findings emphasize the necessity of adjusting hormonal and nutritional circumstances to improve tuber start. The linear impact of these variables was strong, demonstrating a consistent and predictable correlation between the input variables and tuberization efficiency. For tubers per plant, BAP was identified as a significant factor (F = 7.01, P = 0.009), underscoring its influence on enhancing tuber yields. This indicates that elevating BAP concentration may result in a higher quantity of tubers, which is very pertinent for enhancing commercial production. The dimensions and mass of tubers were markedly affected by both IBA and NAA. IBA significantly influenced tuber size (F = 7.99, P = 0.005) and weight (F = 7.79, P = 0.006), while NAA also had a notable impact on tuber size (F = 8.04, P = 0.005) and weight (F = 10.73, P = 0.001). The findings indicate that IBA and NAA are essential for enhancing tuber quality by augmenting both size and weight, offering guidance for the optimization of growth circumstances. The RSM-based regression analysis identified BAP, IBA, NAA, and sucrose as the principal factors influencing tuberization, tuber production, and quality (Table 3).

Results of RSRA were further analyzed by constructed Pareto charts and normal plots. A similar trend for tuberization (Fig 1a) and tubers per plant (Fig 1b) was observed. BAP and sucrose were found as the most significant factors for both parameters while BAP was significant for tuberaization. The analysis of the normal plot illustrated the trend of increased tuberization (Fig 1c) and tubers per plant (Fig 1d) with BAP and sucrose. A similar trend was also observed for tuber size (Fig 1e) and tuber weight (Fig 1f) by Pareto chart, and both factors were significantly regulated by IBA (Fig 1g) and NAA (Fig 1h). Supplementation of both auxins enhanced the tuber size and weight with an increase in their concentration. A comparison of both auxins revealed the greater impact of NAA compared to IBA.

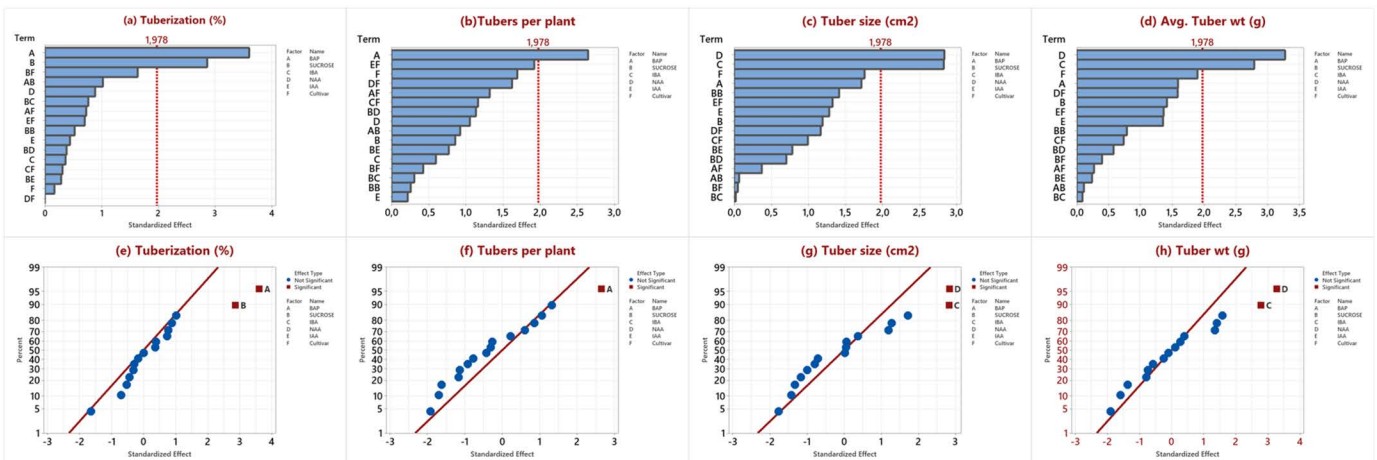

**Fig 1. Pareto chart and normal plot analysis of in vitro tuber formation (a,e) tuberization, (b,f) tubers per plant, (c,g) tuber size, and (d,h) tuber weight.**

## RSRA analysis of cultivars

Results of RSRA did not exhibit the impact of cultivar, and RSRA analysis of both cultivars was performed individually and presented in Table 4. Results revealed the statistical significance of the model and linear analysis for both cultivars. Comparative analysis of BAP for both cultivars revealed the significant impact of BAP on tuberization (p0.038) for Fontana cv. Whereas tuberization (p0.004) and tubers per plants (p0.006) were found statistically significant for Agria cv. All parameters of Agria cv. were statistically affected by sucrose, while no impact was recorded for Fontana cv. Supplementation of IBA regulated the tuber size (p0.021) and tuber weight (p0.034), while NAA affected the tuber weight of Fontana cv. A combination of Sucrose x NAA (p0.040) and Sucrose x IAA exerted a significant impact on the tubers per plant of Fontana cv. Whereas BAP x Sucrose regulated the tubers per plant of Agria cv. All other linear and two-way interactions were found statistically insignificant for both cultivars.

Results of the Pareto chart and normal plot analysis of Agria and Fontana are presented in Figs 2 and 3 respectively. The results of tuberaization of Agria cv. were statistically significant (Fig 2a) and placed on the right side of the line in a normal plot indicating the possibility of enhanced tuberaization with an increase in BAP concentration (Fig 2b). Results of tubers per plant revealed the BD and DE (Fig 2c) as the most significant factor but exerted a negative impact (Fig 2d). For tuber size (Fig 2e, g) and tuber weight (Fig 2f,h), IBA and NAA were the most significant factors and exerted a positive impact. On the contrary, a similar trend was observed for Agria cv. and Pareto chart analysis revealed sucrose as the most significant factor for all parameters and the impact of sucrose revealed an increase with a parallel increase in sucrose concentration (Fig 3a-h).

## Heatmap analysis

Results of heatmap analysis of Fontana cv. (Fig 4a) revealed the strong correlation between all output parameters, and illustrated the dependency on each other and also on input factors. Sucrose was the most significant factor and displayed a positive correlation for tuberization (0.67), tubers per plant (0.32), tuber size (0.32), and tuber weight (0.32). BAP concentration exhibited moderate but positive correlations for all output parameters within the range of 0.28–0.32. Among the auxins, IBA and NAA exhibited a negative correlation with each other (−0.25) and with IAA (−0.25). IAA exhibited weak but positive correlations for all parameters. Results revealed that both sucrose and BAP are highly significant for tuberaization and tuber growth. Results of heatmap analysis of Agria cv. (Fig 4b) almost followed a similar pattern to Fontana cv. Sucrose was found most significant factor and displayed a positive correlation for tuberization (0.49), tuber size (0.45), and tuber weight (0.49). Whereas BAP exhibited moderate correlations for all output parameters with a range of 0.30–0.35. Among the auxins, IBA and NAA exhibited a negative correlation with each other (−0.25) and with IAA (−0.25). However, IBA exhibited a positive but relatively low correlation with tuber size (0.21) and tuber weight (0.22). IAA was found as most least factor and exhibited weak or negative correlations.

## Network plot analysis

Results of network plot analysis confirmed the findings of heatmap analysis of Fontana cv. (Fig 5a), and tuberization regulated the remaining parameters. The connection between sucrose and tuberization is also prominent and exhibits the significance of sucrose concentration. BAP exhibits positive but mild interactions with tuberization and other parameters. The link between NAA-IBA, and IAA with other plant growth regulators was found weak. Results of Agrai cv. also exhibited a similar type of strong relationship between tuberization and other parameters (Fig 5b). BAP hormone was found more influential than auxins for tuber traits. The relationship between all auxins is relatively weak indicating the lesser impact on tuberization.

## Machine learning analysis

Data generated during in vitro regeneration was validated and predicted by three different ML models with the help of five performance metrics. The performance of the models was predicted with a coefficient of determination ($R^2$). Results

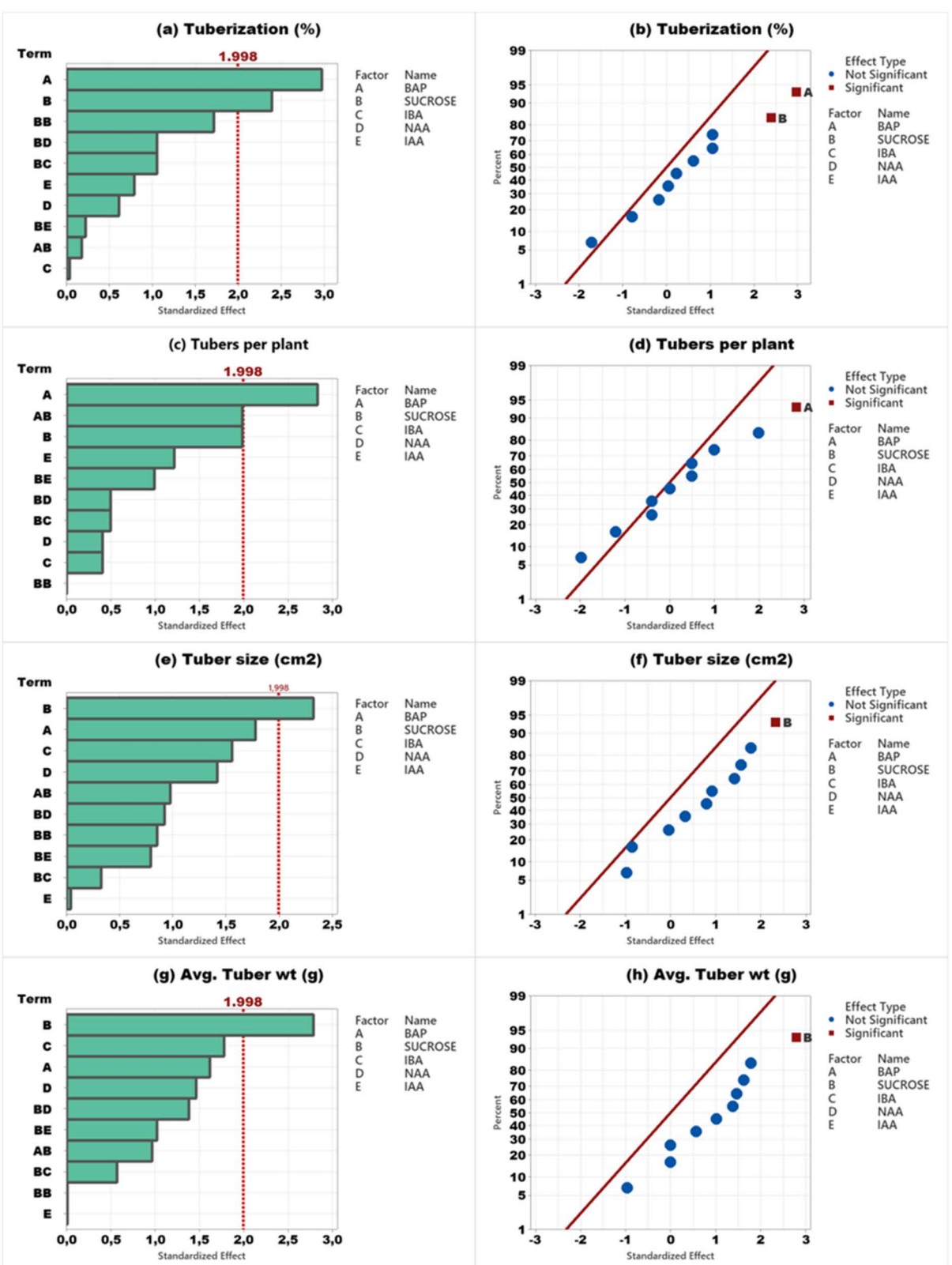

**Fig 2. Pareto chart and normal plot analysis of in vitro tuber formation of Agria cv.** (a,e) tuberization, (b,f) tubers per plant, (c,g) tuber size, and (d,h) tuber weight.

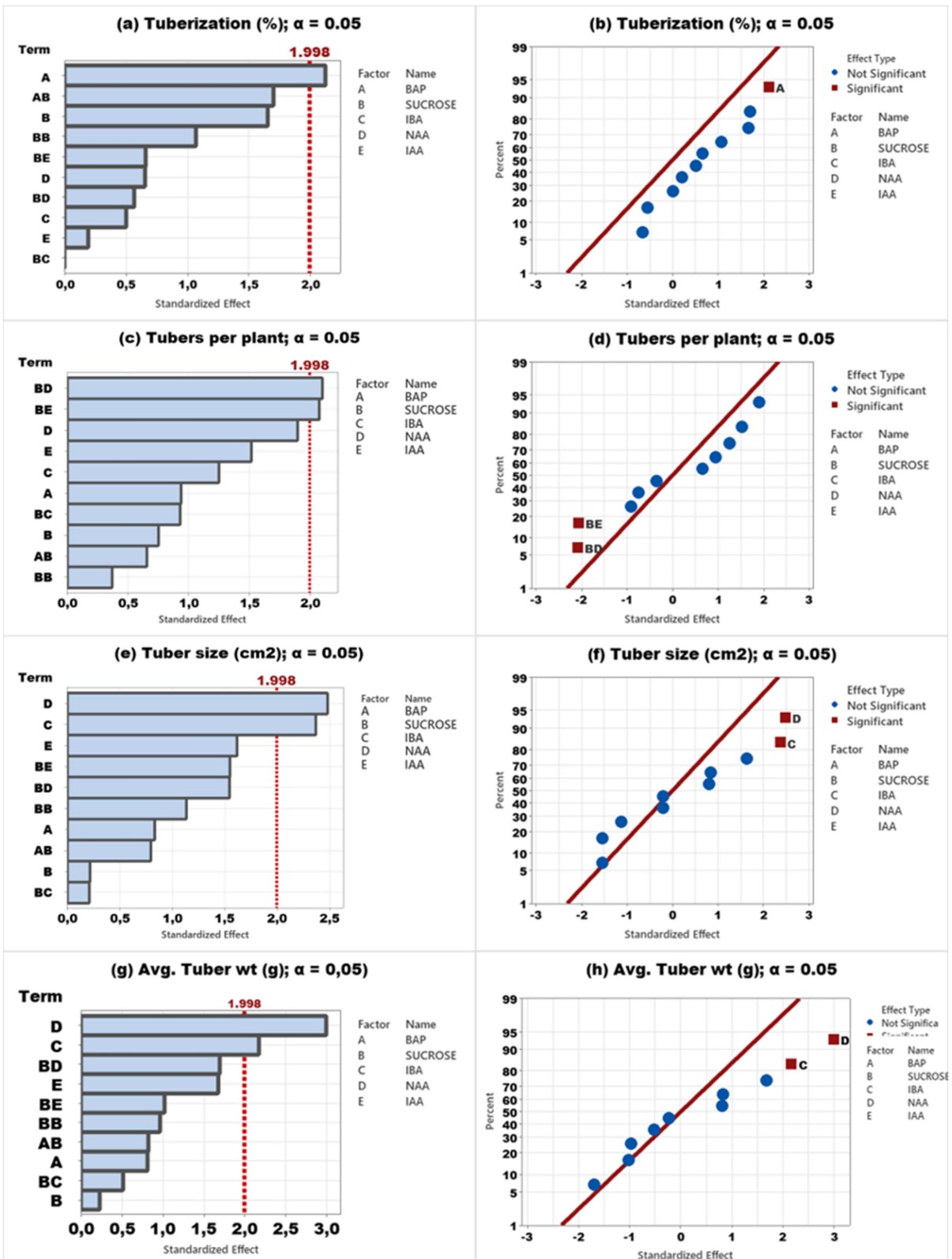

**Fig 3. Pareto chart and normal plot analysis of in vitro tuber formation of Fontana cv.** (a,e) tuberization, (b,f) tubers per plant, (c,g) tuber size, and (d,h) tuber weight.

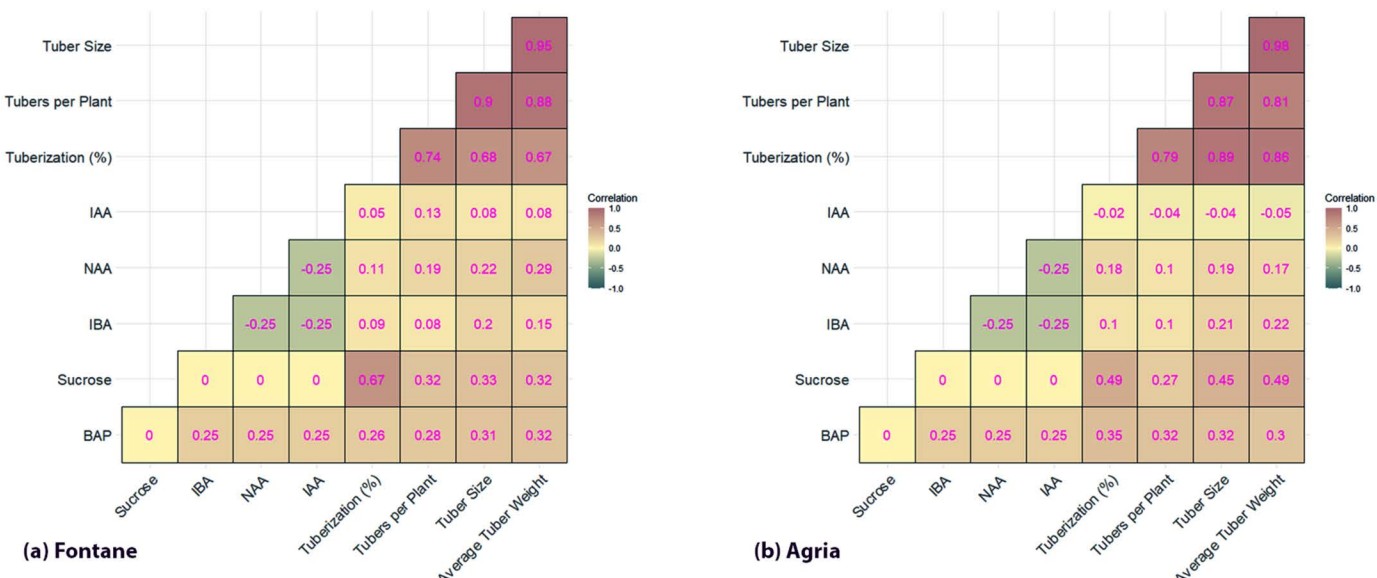

**Fig 4. Heatmap analysis of in vitro tuber formation of potato cultivars (a) Fontane cv.** (b) Agria cv.

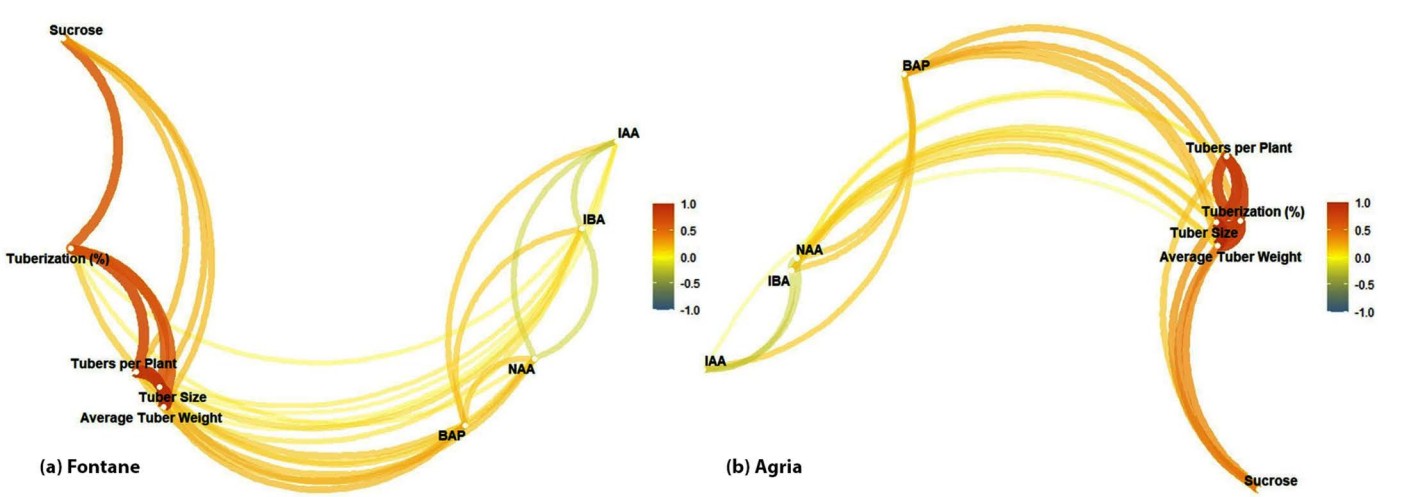

**Fig 5. Network plot analysis of in vitro tuber formation of potato cultivars (a) Fontane cv.** (b) Agria cv.

revealed maximum prediction by $R^2$ score of 0.379 by RF model that was followed by SVR (0.306) and LightGBM (0.256). However, all models exhibited high RMSE scores, indicating that models struggled to minimize errors, and a minimum was recorded for the RF model. The results of MedAE illustrated the minimum score from the LightGBM model (16.535). ML results for tubers per plant were lower than tuberization and the maximum $R^2$ score was computed by SVR (0.125) followed by LightGBM (0.108) and RF (0.083) model. Results of RMSE and MAE were also similar and ranged from 0.460–0.471 and 0.417–0.419 respectively, indicating comparable prediction errors. MLSE scores were also low and recorded in the range of 0.101–0.106, indicating low absolute deviation irrespective of high $R^2$ scores. LightGBM showed the highest MedAE (0.410), suggesting that it had less stability in predicting the number of tubers per plant (Table 5).

**Table 5. Performance metrics for the ML models for in vitro tuberization of potato.**

| Performance Mettric | $R^2$ | RMSE | MAE | MLSE | MedAE |
|---|---|---|---|---|---|
| **RF** | | | | | |
| Tuberization (%) | 0.379 | 28.020 | 22.121 | 3.838 | 17.925 |
| Tubers per plant | 0.083 | 0.471 | 0.419 | 0.106 | 0.380 |
| Tubers size ($cm^2$) | 0.188 | 0.273 | 0.222 | 0.043 | 0.184 |
| Average tuber weight (g) | 0.205 | 0.119 | 0.097 | 0.011 | 0.076 |
| **LightGBM** | | | | | |
| Tuberization (%) | 0.256 | 30.676 | 21.900 | 3.390 | 16.535 |
| Tubers per plant | 0.108 | 0.464 | 0.419 | 0.104 | 0.410 |
| Tubers size ($cm^2$) | 0.197 | 0.271 | 0.220 | 0.043 | 0.166 |
| Average tuber weight (g) | 0.189 | 0.120 | 0.095 | 0.011 | 0.076 |
| **SVR** | | | | | |
| Tuberization (%) | 0.306 | 29.632 | 24.671 | 3.743 | 21.166 |
| Tubers per plant | 0.126 | 0.460 | 0.417 | 0.101 | 0.300 |
| Tubers size ($cm^2$) | 0.187 | 0.273 | 0.239 | 0.043 | 0.227 |
| Average tuber weight (g) | 0.246 | 0.115 | 0.096 | 0.010 | 0.089 |

For tuber size, the LightGBM model exhibited a high $R^2$ score of 0.197 compared to the RF (0.188) and SVR (0.187) models. RMSE (0.271–0.273) and MAE (0.220–0.239) scores reflected the difficulties faced by models to predict accurately. The LightGBM model also generated the lowest MedAE (0.166) while the RF model had the highest score of 0.227. The SVR model had the better predictive ability of the SVR model for tuber weight and computed $R^2$ score of 0.246 followed by RF (0.205) and LightGBM (0.189). The SVR model also yielded a low RMSE (0.115) compared to other models, indicating better predictive accuracy. Although the $R^2$ score of the RF model was less than the SVR model, it also yielded low RMSE (0.119), MAE (0.097), and MLSE (0.011) scores. Similarly, LightGBM also computed a low RMSE score (0.120) irrespective of a low $R^2$ score. A comparison of overall model performance revealed that no model outperformed the other model. RF model exhibited superior performance for tuberization, while the SVR model outperformed other models for tubers per plant and tuber weight. The LighGBM model predicted better accuracy for tuber size. The results of ML analysis are presented in Fig 6a-6d, which confirmed the results of different models. Results illustrated the difference in actual and predicted scores for tuberization (Fig 6a), tubers per plant (Fig 6b), tuber size (Fig 6c), and tuber weight (Fig 6d) due to low predicted scores exhibited by models.

## Discussion

Comparison of varieties based on four key parameters revealed the better performance of Fontana cv. due to genetic variation, environmental interaction, and hormonal sensitivity, which enables the CV to adapt better under in vitro conditions and respond to growth regulators [18]. Results further illustrated the significance of BAP supplementation in the culture media and resulted in an almost 40.0% increase in all parameters compared to the control. Active cell division and differentiation, optimal concentration, and improved nutrient utilization are the possible reasons regulated by the presence of BAP [19]. Supplementation of sucrose concentration exerted significant impacts on in vitro tuber production, and all parameters increased with increased sucrose concentration. Sucrose is the primary energy source in tissue culture and its interaction with growth hormones and facilitating starch synthesis is highly significant for optimization [20].

Investigation of individual factors reveals the importance of sucrose, BAP, and IBA in optimizing in vitro potato tuberization. The optimal combination of 90 g/L sucrose, 2 mg/L BAP, and 1 mg/L IBA resulted in the most favorable medium for potato tuberization. Results revealed the synergistic effects of sucrose, BAP, and IBA for enhanced hormone efficiency,

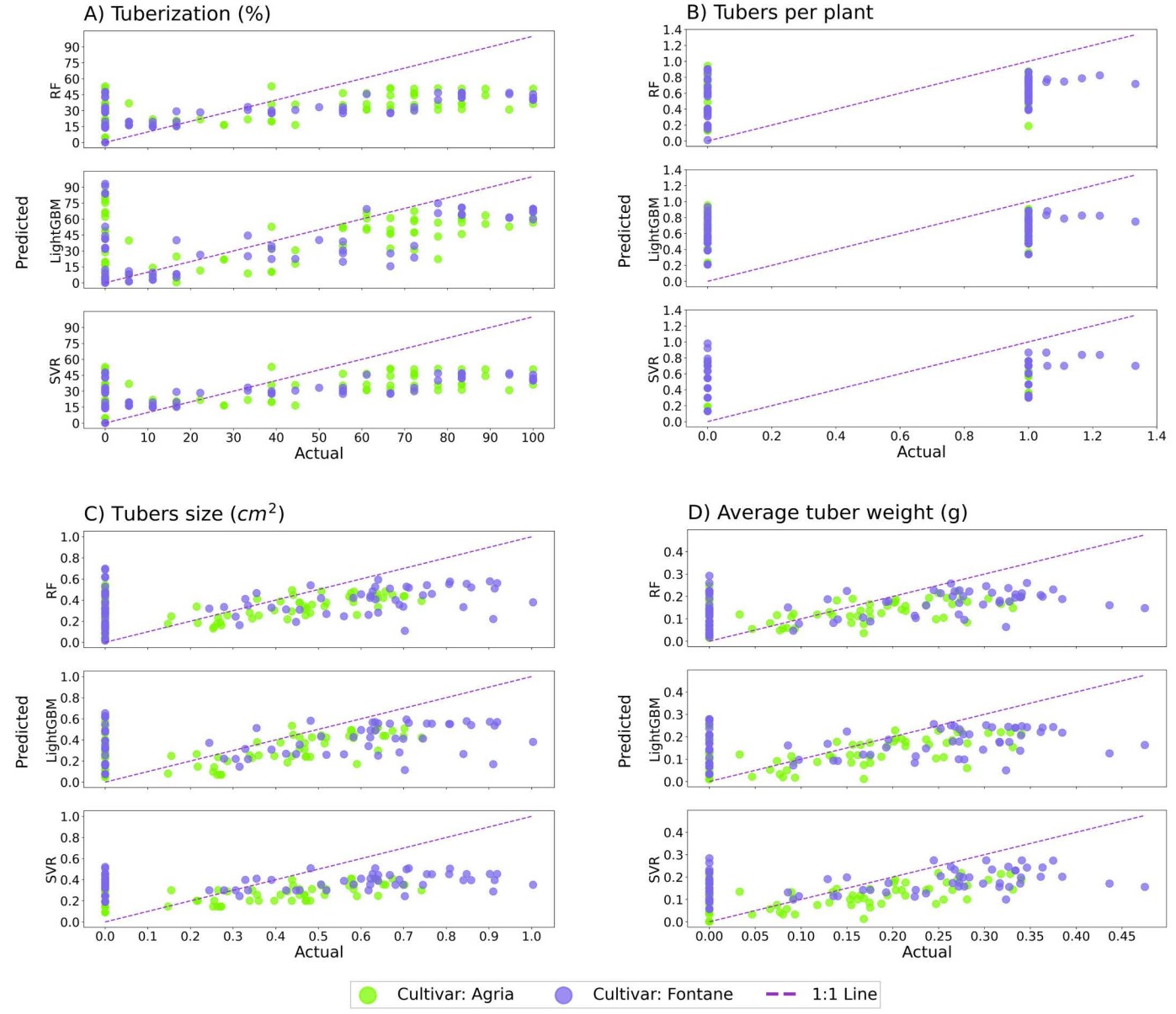

**Fig 6. Machine learning analysis of in vitro tuber formation of potato cultivars (A) tuberization, (B) tubers per plant, (C) tuber size, and (D) tuber weight.**

improved nutrient utilization, and cumulative benefits for potato tuberization [21]. Results significantly emphasized the inclusion of IBA in the culture medium due to enhanced tuber size and weight, possibly due to cell elongation, nutrient translocation, root initiation, differentiation, and tuber growth by IBA. On the other hand, a higher concentration of sucrose likely provided sufficient energy for robust tuberization. Whereas BAP supplementation maintained the hormonal requirements needed for robust tuber formation [22].

It was evident from the results that cytokinin-auxin and sucrose regulated the in vitro tuber formation, but which factor was more critical was not clear. Therefore, RSRA was performed and it revealed the robustness of the model with significant

F-values [23]. Results of linear analysis illustrated the dominancy of sucrose and BAP for tuber formation, While IBA and NAA controlled the tuber size and weight. These results revealed the significance of linear factors for optimization and crucial for experiment or commercial production [24]. Whereas Two-way interaction was not as effective as linear and was found insignificant, indicating minimal synergistic or antagonistic impact between variables and dominant feature of independent variables [25]. Identification of dominant and recessive factors is important for the optimization process and it allows RSM to develop visual charts [26] or plots like Pareto charts, normal plots, contour, and surface plots for the optimization process. Results of Pareto charts and normal plots revealed the most significant and dominant factors for both cultivars.

RSRA analysis of individual cultivars was performed to identify the cultivar-specific responses for in vitro tuberization due to a similar trend exhibited by both cultivars combined. RSRA is an advanced computational model that allows us to understand the impact of linear and two-way interactions of input factors. RSRA analysis provided the factors that regulated the in vitro tuberization of potatoes for both cultivars independently. Such types of analysis are helpful to optimize in vitro regeneration protocol efficiently [18], and confirmed the supremacy of model-driven optimization of previous findings in plant tissue culture for hormonal [20] and sucrose concentration (Ozcan 2023).

Investigation of Pareto charts and normal plots confirmed the findings by computing the dominant factor with their impact on in vitro tuberization. Results illustrated the significance of BAP concentration for Agria cv. [27] possibly due to more cell division and suppression of apical dominance. Results are in line with the previous studies in potato tuberization where high BAP concentration led to improved micro tuber development possibly due to enhanced starch biosynthesis and carbohydrate partitioning [28]. Results of Pareto chart analysis of tubers per plant also revealed the significance of BAP alone for promoting tuberization [29] and no impact was recorded for BAP × sucrose and BAP × IBA. On the contrary, IBA and NAA were found most significant for tuber size and weight due to enhancing cell elongation, root formation, and nutrient translocation by IBA [30], while NAA increases tuber weight and yield [31]. Results of Pareto charts for Fontana cv illustrated the dependency on sucrose for all parameters confirming the results that tuberization can be increased with elevated sucrose concentration.

Heatmap analysis is a data visualization tool used to identify key growth factors and their interactions in plant sciences [32]. In this study, heatmap helped to understand the positive or negative correlation between PGRsand sucrose with tuber formation and biomass parameters [33]. Results revealed a strong correlation between tuberization traits and controlling factor and sucrose was the most significant factor followed by BAP for Fontana cv. Whereas heatmap analysis confirmed the negative impact of auxins [34]. A similar type of correlation was observed for Agria cv. like Fontana cv. IBA was found essential for tuber size and weight but its impact was relatively weak. Results of the heatmap revealed key insights for optimizing in vitro tuberization of potatoes. It can be concluded that optimization of sucrose and BAP along with adjusting auxins type or concentration are critical for potato tuberization.

Results were further confirmed by network plot analysis that is used to investigate complex interactions between biological parameters in plant sciences. It helps to identify the interdependence of different factors, and direct and indirect influences [35]. Results illustrate the role of PGRs and sucrose and confirm the findings of heatmap analysis [36]. A weak connection between NAA and IBA, and between IAA and other PGRs revealed the minimum impact of auxins on tuber formation of Fontana cv. On the contrary, BAP was the most significant factor for Agria cv. and exhibited the positive impact of cytokinins for tuber formation in genotypes with a lower endogenous cytokinin reserve [37]. It can be concluded from the network plot analysis that sucrose is the most dominant factor followed by BAP, but BAP is also dependent on sucrose. Whereas auxins exhibited a secondary role in in vitro tuber induction. Both visual tools not only confirmed statistical outputs but also highlighted central regulatory variables such as sucrose and BAP. The heatmap revealed strong positive correlations, while the network plot emphasized trait interdependencies, allowing a clearer understanding of how individual factors influenced multiple tuberization traits.

Results of ML models revealed that the model's prediction ability was associated with the specific parameters. Among the models, the RF model exhibited better prediction ability for tuberization and successfully indicated the variation in

the data set compared to other models [38]. However, the prediction was low due to significant errors exhibited by other metrics [39]. Although better performance of the RF model has been documented in plant sciences [40], overfitting and high error scores in complex biological data are the possible reasons for low prediction by the RF model [41]. The SVR model exhibited better predictive ability for tubers per plant and tuber weight, but it exhibited relatively high MAE scores, indicating its limitations in handling non-linear relationships, especially in plant sciences [42]. Whereas LightGBM performed other models for tuber size and exhibited the lowest MedAE score along with a low $R^2$ score. It can be extracted from the results that the LightGBM model exhibited stable median prediction with fewer extreme errors [43]. Whereas low $R^2$ scores by SVR models suggested that input factors like plant growth regulators and other factors may regulate the tubers per plant. It can be concluded from the results that none of the models are efficient for all parameters and failed to minimize absolute errors due to complex genetic and environmental factors regulating the whole in vitro regeneration process [44]. Results revealed the significance of other factors and highlighted the need for further feature engineering in ML models. Application of ensemble learning approaches or hybrid modeling to improve predictive performance, as linear and non-linear models struggled equally in terms of RMSE and MAE scores [40]. It is recommended that future research should focus on improving feature selection strategies, using genetic algorithms or PCA, to optimize predictor variables for better model performance [38]. Incorporating real-time monitoring data and expanding the training dataset size could enhance the robustness and generalizability of predictions [45].

## Conclusion

This study successfully demonstrated the potential of integrating RSM with AI techniques to optimize in vitro potato tuberization. The results emphasized the critical roles of sucrose and BAP, the complex interactions of auxins, and the varying responses between cultivars. The integration of advanced statistical and computational tools offers a promising pathway for improving tissue culture efficiency, ultimately contributing to more sustainable and productive potato cultivation practices. The results have significant implications for both research and commercial potato production, as fine-tuning sucrose concentration, cytokinin-auxin balance, and leveraging RSM and AI-driven models can significantly improve the efficiency of in vitro tuberization. Future research should focus on incorporating real-time monitoring of culture conditions, exploring additional machine-learning techniques, and expanding the experimental design to include different potato genotypes and environmental variables.

## Author contributions

**Conceptualization:** Rajermani Thinakaran, Muhammad Aasim.

**Data curation:** Ecenur Korkmaz, Başak Ünver.

**Formal analysis:** Ecenur Korkmaz, Seyid Amjad Ali, Zeshan Iqbal.

**Investigation:** Ecenur Korkmaz, Başak Ünver, Muhammad Aasim.

**Methodology:** Seyid Amjad Ali, Muhammad Aasim.

**Project administration:** Rajermani Thinakaran, Zeshan Iqbal.

**Resources:** Ecenur Korkmaz, Başak Ünver.

**Software:** Seyid Amjad Ali, Zeshan Iqbal, Muhammad Aasim.

**Validation:** Seyid Amjad Ali.

**Visualization:** Rajermani Thinakaran, Seyid Amjad Ali, Muhammad Aasim.

**Writing – original draft:** Muhammad Aasim.

**Writing – review & editing:** Rajermani Thinakaran, Ecenur Korkmaz, Başak Ünver, Seyid Amjad Ali, Zeshan Iqbal.

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
