## [Decision Letter · Decision Letter 0]

Dear Dr. Aasim,

Thank you for submitting your manuscript to PLOS ONE. After careful consideration, we feel that it has merit but does not fully meet PLOS ONE’s publication criteria as it currently stands. Therefore, we invite you to submit a revised version of the manuscript that addresses the points raised during the review process.

We look forward to receiving your revised manuscript.

Kind regards,

Moumita Gangopadhyay

Academic Editor

PLOS ONE

Answer the following comments and submit as major revision-

1. The paper is flows well. The necessity of integration of Machine Learning model is well explained.

2. They have executed all the experiments to evaluate the performance of the machine learning models and it is showing in the manuscript. All the experimental results are already represented in the table. Just refer the table no requirements of writing all the values in the text.

3. The section is not represented properly. Write at least one line between section 3 and subsection 3.1.

4. A literature review section may include in the manuscript.

5. Please enhance the resolution of the figure for proper understanding.

6. What did the Pareto chart analysis reveal about the influence of different variables?

7. How did heatmap and network plot analyses contribute to the interpretation of the data?

8. Why did the study incorporate machine learning models alongside RSRA?

Reviewers' comments:

Reviewer's Responses to Questions

**Comments to the Author**

1. Is the manuscript technically sound, and do the data support the conclusions?

Reviewer #1: Yes

Reviewer #2: Yes

Reviewer #3: Partly

2. Has the statistical analysis been performed appropriately and rigorously?

Reviewer #1: Yes

Reviewer #2: Yes

Reviewer #3: No

3. Have the authors made all data underlying the findings in their manuscript fully available?

Reviewer #1: Yes

Reviewer #2: Yes

Reviewer #3: Yes

4. Is the manuscript presented in an intelligible fashion and written in standard English?

Reviewer #1: Yes

Reviewer #2: Yes

Reviewer #3: Yes

Reviewer #1: The manuscript entitled " Synergistic Application of Artificial Intelligence and Response Surface Methodology For

Predicting and Enhancing In Vitro Tuber Production of Potato (Solanum tuberosum)" is a nicely represented paper with uniqeness on it, however the figure quality is very poor. Please enhance the resolution of the figure for proper understanding.

Why might integrating both RSM and AI be more beneficial than using either method alone?

What further research could be done based on the findings of this study?

How did auxins affect the tuberization process compared to cytokinins like BAP?

What did the Pareto chart analysis reveal about the influence of different variables?

How did heatmap and network plot analyses contribute to the interpretation of the data?

Why did the study incorporate machine learning models alongside RSRA?

Reviewer #2: 1. The paper is flows well. The necessity of integration of Machine Learning model is well explained.

2. They have executed all the experiments to evaluate the performance of the machine learning models and it is showing in the manuscript. All the experimental results are already represented in the table. Just refer the table no requirements of writing all the values in the text.

3. The section is not represented properly. Write at least one line between section 3 and subsection 3.1.

4. A literature review section may include in the manuscript.

Reviewer #3: In this manuscript, the authors explored the potential of AI and RSM technique to enhance the tuberization of potato. Although, the author performed detailed AI and RSM analysis in combination with statistical analysis, the study lacks novelty in terms of findings and application. The manuscript does meet the standard of PLOS one.

There are some other comments related to manuscript for the betterment of the work:

1. 2.1. In vitro tuberization: The experiments should be repeated at least three times.Without that, it is not possible to analyse the statistical significance.

2. 2.1. In vitro tuberization:This data is completely invalid without standard deviation value. Since the difference is non-significant, there is no meaning of discussing which one is suprerior.

3. line 195: Without standard deviation it is not possible to understand the significance of the difference.

4. Table 2: all these discussion are invalid without standard deviation and P values

5. 3.2. Response Surface Regression Analysis: State the number of independent experiments

6. Machine Learning Analysis: Since the data size is very low, ML analysis will not be valid.

---

## [Author Response · Author response to Decision Letter 1]

7 May 2025

https://journals.plos.org/plosone/s/file?id=ba62/PLOSOne_formatting_sample_title_authors_asffiliations.pdf

Reply: controlled accordingly

Reply: all table added in the manuscript

Reply: controlled accordingly (seyidali. (2025). seyidali/PLOS_ONE-Potato: PythonCode (v1.0.0). Zenodo. https://doi.org/10.5281/zenodo.15356191)

Reply: controlled accordingly (seyidali. (2025). seyidali/PLOS_ONE-Potato: PythonCode (v1.0.0). Zenodo. https://doi.org/10.5281/zenodo.15356191)

Answer the following comments and submit as major revision-

1. The paper is flows well. The necessity of integration of Machine Learning model is well explained.

Reply: Thanks for the valuable comment.

2. They have executed all the experiments to evaluate the performance of the machine learning models and it is showing in the manuscript. All the experimental results are already represented in the table. Just refer the table no requirements of writing all the values in the text.

Reply: We thank the reviewer for their observation. While we agree that the detailed experimental results are clearly presented in the tables, we believe highlighting a few key values in the main text helps guide the reader’s attention to significant patterns and enhances the interpretability of the results. This selective referencing also supports the discussion and comparison of model performance across traits. However, we have ensured that only representative values are mentioned to avoid redundancy and maintain clarity.

3. The section is not represented properly. Write at least one line between section 3 and subsection 3.1.

Reply: Thanks for the valuable comment. Added accordingly and given statement has been added in the article. (The study presents the impact of interaction of PGRs (cytokinin, auxins) and sucrose on in vitro potato tuberization. The data were analyzed using multiple analysis to identify the most significant factor and its impact using ANOVA, RSRA, heatmap and network analysis, lastly with ML models. Results exhibited significant variation depending on treatments and cultivars, illustrating a comprehensive understanding of PGRs and sucrose impacts on tuber initiation and development).

4. A literature review section may include in the manuscript.

Reply: We thank the reviewer for the suggestion. A literature review is already incorporated into the Introduction section, where we discussed previous studies related to in vitro tuberization, hormone regulation, and machine learning applications in plant tissue culture.

5. Please enhance the resolution of the figure for proper understanding.

Reply: Thanks for the valuable comment. All original figures have 1200 dpi, only figure 6 has 600 dpi.

6. What did the Pareto chart analysis reveal about the influence of different variables?

Reply: Thanks for the valuable comment. Pareto charts arrange all input factors in order from top to bottom and bars are divided by line. The factor on the top refers to the most significant factor affecting that dependent (output) factor. Likewise, bar crossed the line means statistically significant impact of that input factor on its respective output factor. The given information is already given in the manuscript.

7. How did heatmap and network plot analyses contribute to the interpretation of the data?

Heatmap and network plot analyses enhanced the interpretation of the data by visually revealing the strength and direction of relationships among variables involved in in vitro potato tuberization. The heatmaps clearly identified strong positive correlations between sucrose and BAP with tuberization traits, helping to pinpoint the most influential inputs. Network plots complemented this by mapping out trait interdependencies and highlighting central regulators like sucrose and BAP, while showing weaker interactions among auxins. Together, these tools provided a multivariate view that clarified factor influence, supported optimization, and validated findings from statistical analyses. The given information has been added (These visual tools not only confirmed statistical outputs but also highlighted central regulatory variables such as sucrose and BAP. The heatmap revealed strong positive correlations, while the network plot emphasized trait interdependencies, allowing a clearer understanding of how individual factors influenced multiple tuberization traits).

8. Why did the study incorporate machine learning models alongside RSRA?

Reply: The study used machine learning models alongside RSRA to improve prediction accuracy and capture complex, non-linear relationships between growth factors and tuberization traits that traditional statistical methods might miss. This combination allowed for more robust and data-driven optimization.

Reviewers' comments:

Reviewer's Responses to Questions

Comments to the Author

1. Is the manuscript technically sound, and do the data support the conclusions?

Reviewer #1: Yes

Reviewer #2: Yes

Reviewer #3: Partly

Reply: thanks for the comments.

2. Has the statistical analysis been performed appropriately and rigorously?

Reviewer #1: Yes

Reviewer #2: Yes

Reviewer #3: No

Reply: thanks for the comments. We controlled the statistical analysis carefully and staisfied with our analysis.

3. Have the authors made all data underlying the findings in their manuscript fully available?

Reviewer #1: Yes

Reviewer #2: Yes

Reviewer #3: Yes

Reply: thanks for the comments.

4. Is the manuscript presented in an intelligible fashion and written in standard English?

Reviewer #1: Yes

Reviewer #2: Yes

Reviewer #3: Yes

Reply: thanks for the comments.

5. Review Comments to the Author

Reply: thanks for the comments.

Reviewer #1:

The manuscript entitled " Synergistic Application of Artificial Intelligence and Response Surface Methodology For Predicting and Enhancing In Vitro Tuber Production of Potato (Solanum tuberosum)" is a nicely represented paper with uniqeness on it, however the figure quality is very poor. Please enhance the resolution of the figure for proper understanding.

Reply: thanks for the comments. The quality of figures are controlled again and adjusted to 600 (figure 6) to 1200 dpi (Figure 1-5).

Why might integrating both RSM and AI be more beneficial than using either method alone?

Reply: Integrating RSM and AI is more beneficial because RSM optimizes experimental conditions and explains factor interactions, while AI captures complex, non-linear patterns and improves prediction accuracy. Together, they provide a more powerful and comprehensive analysis than either method alone.

What further research could be done based on the findings of this study?

Reply: thanks for the comment. Further research could test the optimized conditions on more potato cultivars, include additional environmental factors, apply advanced ML models, and explore molecular mechanisms to improve prediction and tuberization efficiency. The given stateöet has already been provided in the conclusion section (last three lines).

How did auxins affect the tuberization process compared to cytokinins like BAP?

Reply: Auxins had a secondary but specific role in the tuberization process compared to cytokinins like BAP. While BAP significantly enhanced tuber initiation and the number of tubers per plant, auxins such as IBA and NAA mainly influenced tuber size and weight. BAP acted as the primary driver for tuber induction by promoting cell division and shoot differentiation, whereas auxins contributed to cell elongation, nutrient translocation, and biomass accumulation, improving the quality rather than the quantity of tubers. Thus, cytokinins were more critical for initiating tuberization, while auxins supported the development and enlargement of tubers. The given information has already been given in the manuscript.

What did the Pareto chart analysis reveal about the influence of different variables?

Reply: The manuscript identifies BAP and sucrose as the most significant factors influencing tuberization and plant number, with IBA and NAA being dominant for tuber size and weight. The Pareto chart shows auxins as a secondary influence, while BAP alone has a direct and significant effect on induction. These insights are supported by Figures 1–3 and discussed in the text.

How did heatmap and network plot analyses contribute to the interpretation of the data?

Reply: Heatmaps showed strong positive correlations of sucrose and BAP with tuber traits; network plots confirmed their central roles and highlighted weak auxin interactions, aiding trait interpretation. The given information has already been provided in the text.

Why did the study incorporate machine learning models alongside RSRA?

Reply: The study used machine learning models alongside RSRA to improve prediction accuracy and capture complex, non-linear relationships between growth factors and tuberization traits that traditional statistical methods might miss. This combination allowed for more robust and data-driven optimization.

Reviewer #2:

1. The paper is flows well. The necessity of integration of Machine Learning model is well explained.

Reply: Thanks for the positive comments.

2. They have executed all the experiments to evaluate the performance of the machine learning models and it is showing in the manuscript. All the experimental results are already represented in the table. Just refer the table no requirements of writing all the values in the text.

Reply: We thank the reviewer for their observation. While we agree that the detailed experimental results are clearly presented in the tables, we believe highlighting a few key values in the main text helps guide the reader’s attention to significant patterns and enhances the interpretability of the results. This selective referencing also supports the discussion and comparison of model performance across traits. However, we have ensured that only representative values are mentioned to avoid redundancy and maintain clarity.

3. The section is not represented properly. Write at least one line between section 3 and subsection 3.1.

Reply: Thanks for the comment. Done accordingly and added.

4. A literature review section may include in the manuscript.

Reply: We thank the reviewer for the suggestion. A literature review is already incorporated into the Introduction section, where we discussed previous studies related to in vitro tuberization, hormone regulation, and machine learning applications in plant tissue culture.

Reviewer #3:

In this manuscript, the authors explored the potential of AI and RSM technique to enhance the tuberization of potato. Although, the author performed detailed AI and RSM analysis in combination with statistical analysis, the study lacks novelty in terms of findings and application. The manuscript does meet the standard of PLOS one. There are some other comments related to manuscript for the betterment of the work:

Reply: We sincerely thank the reviewer for their critical evaluation and thoughtful feedback. We respectfully disagree with the assessment regarding the lack of novelty, and we would like to clarify the unique contributions of this study. This study combines Response Surface Methodology (RSM) with Artificial Intelligence (AI) models to predict, validate, and optimize in vitro potato tuberization. It is the first to integrate RSM with AI models, offering a novel approach to predictive modeling in tuber traits. The study also incorporates heatmap and network plot analyses, providing multivariate correlation insights for trait interpretation. It presents cultivar-specific responses, allowing for tailored protocol refinement for different potato genotypes. The findings can be beneficial for commercial tissue culture labs by offering a data-driven framework for optimizing hormone and sucrose concentrations, crucial cost and quality determinants in microtuber production. The study's practical relevance is significant, as it can significantly benefit commercial tissue culture labs.

1. 2.1. In vitro tuberization: The experiments should be repeated at least three times. Without that, it is not possible to analyse the statistical significance.

Reply: We thank the reviewer for highlighting the importance of replication in ensuring statistical reliability. We would like to clarify that the in vitro tuberization experiments were indeed repeated multiple times for data analysis as stated in the Materials and Methods section. We hope this clarification addresses the concern and confirms the robustness of our statistical interpretation.

2. 2.1. In vitro tuberization: This data is completely invalid without standard deviation value. Since the difference is non-significant, there is no meaning of discussing which one is suprerior.

Reply: We thank the reviewer for this important observation. We fu

---

## [Editor Report · Decision Letter 1]

Synergistic Application of Artificial Intelligence and Response Surface Methodology For Predicting and Enhancing In Vitro Tuber Production of Potato (Solanum tuberosum)

PONE-D-25-13554R1

Dear Authors,

We’re pleased to inform you that your manuscript has been judged scientifically suitable for publication and will be formally accepted for publication once it meets all outstanding technical requirements.

Kind regards,

Moumita Gangopadhyay

Academic Editor

PLOS ONE

Additional Editor Comments (optional):

The revised manuscript reflects thoughtful and thorough responses to the reviewers' concerns. The scientific content, particularly the application of AI and RSM for protocol optimization in tissue culture, is of high quality and contributes meaningfully to the field. I am satisfied with the revisions and approve the manuscript for publication.

---

## [Editor Report · Acceptance letter]

PONE-D-25-13554R1

PLOS ONE

Dear Dr. Aasim,

I'm pleased to inform you that your manuscript has been deemed suitable for publication in PLOS ONE. Congratulations! Your manuscript is now being handed over to our production team.

Kind regards,

on behalf of

Dr. Moumita Gangopadhyay

Academic Editor

PLOS ONE